# Xylan Hemicellulose: A Renewable Material with Potential Properties for Food Packaging Applications

**Petronela Nechita** [1,*] , **Roman Mirela** [2] **and Florin Ciolacu** [3]

1   Department of Environmental, Applied Engineering and Agriculture, "Dunarea de Jos" University of Galati, 817112 Braila, Romania
2   Doctoral School of Fundamental and Engineering Sciences, "Dunarea de Jos" University of Galati, 817112 Braila, Romania; mirela.roman@ugal.ro
3   Department of Natural and Synthetic Polymers, "Gheorghe Asachi" Technical University of Iasi, 700050 Iasi, Romania; fciolacu@ch.tuiasi.ro
*   Correspondence: petronela.nechita@ugal.ro; Tel.: +40-744704928

**Abstract:** Xylan hemicelluloses are considered the second most abundant class of polysaccharides after cellulose which has good natural barrier properties necessary for foods packaging papers and films. Xylan exists today as a natural polymer, but its utilisation in packaging applications is limited and not sufficiently analysed. In this study, the performances of hardwood xylan hemicellulose in forming uniform films and as biopolymer for paper coatings were analysed. The xylan-coated paper and film samples were tested regarding their water, air, and water vapour permeability, water solubility, mechanical strength, and antimicrobial activity against pathogenic bacteria. Structural analyses of xylan hemicelluloses emphasised a high number of hydroxyl groups with high water affinity. This affects the functional properties of xylan-coated papers but can facilitate the chemical modification of xylan in order to improve their hydrophobic properties and extend their areas of application. The obtained results unveil a promising starting point for using this material in food packaging applications as a competitive and sustainable alternative to petroleum-based polymers.

**Keywords:** xylan hemicellulose; foods packaging; water barrier properties; coated papers; xylan films

## 1. Introduction

Nowadays, the packaging industry is under great pressure from both authorities and consumers to improve its environmental and sustainability credentials in the coming years. Therefore, there is a need for the development of a new generation of packaging that has at least the same technical performances as today's mono- or multi-materials but with demonstrated recyclability and biodegradability. The new EU Plastics Strategy (2018) and German Packaging Law (2019) are intended to counteract the massive increase in the production of plastics worldwide [1–3]. In this context, there are real business opportunities for packaging materials that must be with easy integration into an existing waste value chain, having a high recycling rate, and being biodegradable in compost medium and in marine environments.

Bioresources represent an important vision of a circular economy that explores the importance of research and advancements with regard to the conversion of biological raw materials in the development of innovative value chains. Bioeconomy concept depends on biomass availability, which has an important role in biobased production.

According to this, the utilisation of bioresources from lignocellulose biomass (cellulose, hemicellulose, and lignin) to design and produce bio-based sustainable materials has a high potential for development in the coming years, to replace the oil-based materials [4]. The worldwide production of lignocellulosic biomass is estimated to be about 60 billion tons/year [5].

Hemicelluloses are the second-most abundant class of biopolymers found in plant biomass, after cellulose, and they represent an important renewable resource of biopolymers which, until a few years ago, was usually removed from lignocellulosic biomass with lignin during pretreatment and used by conversion into chemicals, fuel, and as a source of heat energy. For example, in the pulp and paper industry, most hemicelluloses remain in the fibres, to improve the strength properties of paper products or are discarded as waste material during bleaching and other operations. Due to their structural varieties and diversity, hemicelluloses can be utilised for value-added applications in native or modified forms in various areas, including packaging applications [4]. There are many renewable resources of biomass which were insufficiently exploited to obtain hemicelluloses and value-added products. The primary and secondary cell walls of wood and annual plants (cereal straws) contain about 20–35% hemicelluloses; the amounts vary as a function of biomass source, such as hardwood (40%), softwood (35%), cornstalk (31%), maize steins (28%), barley straw (38%), wheat straw (32%), rice straw (24%), rye straw (36%), various agricultural residues (30%), and green algae (50%) [6–9]. When compared with cellulose which occurs in the cell walls as microfibrils, hemicelluloses exist in the matrix phase of cell walls.

In the last two decades, studies have shown a greater application potential of hemicelluloses, emphasised many times by leading polysaccharide scientists, but has not yet been exploited on an industrial and commercial scale [10,11].

Based on their excellent biodegradability, biocompatibility, and bioactivity, HCs and their derivatives have currently received considerable attention in terms of material applications such as edible coatings and films or coatings for paper and board for the packaging industry, hydrogels and binders for medicine or drug delivery and release in the pharmaceutical field, functional composites for heavy metals, and dye adsorption in wastewaters treatment and textile industry, as well as for obtaining biofuels [12–17].

Hemicelluloses consist of various different sugar monomer units arranged in different proportions which include five-carbon sugars and six-carbon sugars called pentoses $(C_5H_8O_4)_n$ and hexoses $(C_6H_{10}O_5)_n$, respectively. Xylose and arabinose are representative sugars units for pentoses and mannose and galactose for hexosanes. In addition to these regular sugars, acidified forms also exist in hemicelluloses, for instance, glucuronic acid and galacturonic acid [4,17].

Hemicelluloses (xylan and glucomannan) are generally extracted from the primary and secondary plant cell walls or as co-products from several industrial processes that use wood as raw material, such as dissolving pulp manufacturing, nanocrystalline cellulose and nanofibrillated cellulose production, or sugar for biofuels [18]. The most popular processes used to extract hemicelluloses are alkaline and hot water extractions, steam extraction, organic solvent, ionic liquid, as well as their combination [19–24].

Depending on the source of plant biomass, different hemicellulose compositions, chemical structures, and amounts could be obtained. For example, the dominant hemicellulose in hardwood is xylan (glucuronoxylan), while in softwood, it is mainly mannan (glucomannans and galactoglucomanns). After cellulose, xylan hemicelluloses exist in large quantities in lignocellulosic biomass, which is the main focus of this paper.

*Xylan: A Major Type of Hemicelluloses*

Xylan polysaccharides are the most abundant hemicelluloses component of hardwood, available in large quantities in secondary cell walls of agro-residues (wheat straw, corn stalks, and cobs) or as secondary products in the wood or pulp and paper industry [4,16,25]. A new source of xylan biopolymers includes some types of seaweed (green and red algae) [25]. In biomass, xylans can be found with various structures based on their botanical source or tissue type. In annual plants and algae, xylans are heteropolymers based on a 1,4-β-D-xylopyranose backbone, which is branched by short carbohydrate chains. However, certain green and red algae contain 1,3-β-D-xylans or 1,3;1,4-β-D-xylans, which are homoxylans [4,26].

The most amount of xylan is generated in the manufacturing process of dissolving pulp, with high α-cellulose content. The existing trends in the increasing demand for dissolving pulps will contribute to a noticeable increase in xylan production and availability in the next few years.

Depending on biomass source and used extraction method, the yield of xylan extraction from hardwood pulps is about 31–67% of the original xylan content [27].

Until now, the commercial applications of xylan hemicelluloses have been limited to xylitol and biofuels obtained from the biological conversion of sugar, starch, and vegetable oils. Xylitol is the most widely produced compound from xylan, and it is obtained by chemical and biotechnical methods with large applications such as food sweeteners for diabetic patients or preventive additives against dental caries in toothpastes composition. Due to their potential prebiotic properties, the xylooligosaccharides produced from xylan are used as dietary fibres or functional foods [28].

Existing in high diversity and complexity, xylans are considered nowadays valuable by-products which can be used as sources of raw materials for different applications, even if there are not available in their entirety for industrial production. Based on the existing studies, the xylan hemicelluloses represent a promising renewable biopolymeric resource for different industrial applications [29].

Generally, xylan hemicelluloses are hydrophilic polymers with extensive hydrogen bonds that limit the area of their industrial applications (Figure 1). The abundance of free hydroxyl groups distributed along the backbone and side chains make it an ideal candidate for chemical functionalisation using a variety of chemical reactions. As result, a new material with appropriate properties, such as hydrophobicity, thermal formability, and the ability for forming films can be obtained. The last feature is necessary for xylan hemicellulose to improve self-supporting barrier films when used in food packaging. This will broaden the applications areas of hemicelluloses [23]. Modified xylans have the potential for wide applications in medicine, as hydrogels and drug delivery, as emulsifying additives in the food industry, or as wet-end additives, coatings, and films in packaging [30]. Literature reviews indicate that the main application of the xylan hemicelluloses is as a barrier material for stand-alone packaging films. Xylan-based films have demonstrated good gas barrier properties against oxygen, grease, and aroma and therefore have the potential for application on food paper packaging, especially in oxygen-sensitive dairy products, greasy snacks, or pet foods, as well as aromatic products such as spices and coffee. Furthermore, coated on the inside of paper packaging, it can prevent the migration of mineral oils from recycled fibres or printing inks into the packaged products [31,32].

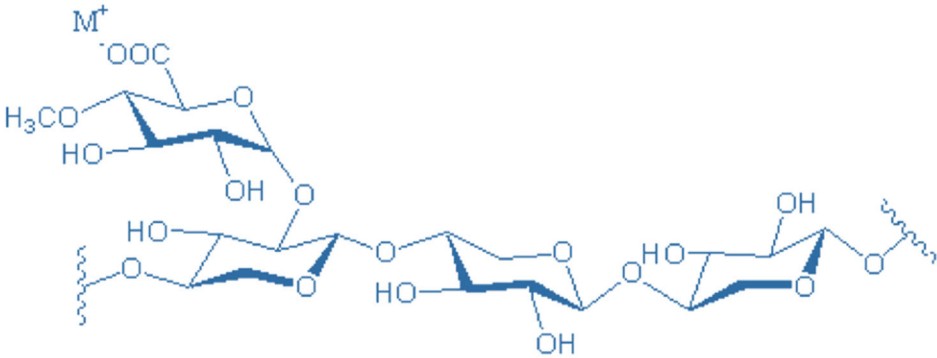

**Figure 1.** Chemical structure of hardwood xylan edited according with [33].

This research study aimed to evaluate the performances of hardwood xylan hemicellulose to form uniform films and as a biopolymer for paper coatings. The samples of films and coated papers were evaluated in terms of mechanical strength, barrier ability to water and water vapour, air permeability, and antimicrobial properties.

## 2. Materials and Methods

### 2.1. Materials

- Commercial *base paper* from unbleached cellulose pulp with a grammage of 50 g/m$^2$ and water absorption, Cobb index of 28 g/m$^2$;
- Xylan hemicellulose from beechwood—brown powder purchased from Carl Roth Company, Germany; it was used as an aqueous dispersion of 20 g/L;
- Chitosan, as fine particulate material, was purchased from the Vanson Company, with high molecular mass and degree of acetylation about 20.8%; it was used as dispersed in acetic acid (1%);
- Glycerol (plasticiser), as a pharmaceutical commercial product with 99% purity.

### 2.2. Methods

- Xylan Hemicellulose's Characterisation

The FTIR spectra were recorded with a solvent-free Bruker Invenio spectrometer, with a horizontal attenuated reflective device (ATR) equipped with diamond crystal, on a spectral window between 400 and 4000 cm$^{-1}$, scan no. = 32 (sample/background), and a resolution of 4 cm$^{-1}$. No prior preparation of samples was required for spectrum recording. A computer, Model OPUS 8.2.8, was used for spectrum recording.

- Film Preparation

Four series of films (Table 1) were prepared by the casting method. Xylan, plasticiser, and chitosan were dispersed in deionised water (1%) and ethanol under magnetic stirring at 1500 rpm and room temperature for 3 h; subsequently, the mixture was cast in a Petri dish (11 cm in diameter) and dried at the laboratory temperature for 4 days (Figure 2).

**Table 1.** Codification of xylan film samples.

| Composition | Sample Code |
|---|---|
| Xylan | XHc |
| Xylan and glycerol (25%) | XG |
| Xylan with ethanol solvent | XP1/25 |
| Xylan:chitosan ratio 50:50 | 50XCh |
| Xylan:chitosan ratio 70:30 | 70XCh |
| Xylan:chitosan ratio 80:20 | 80XCh |

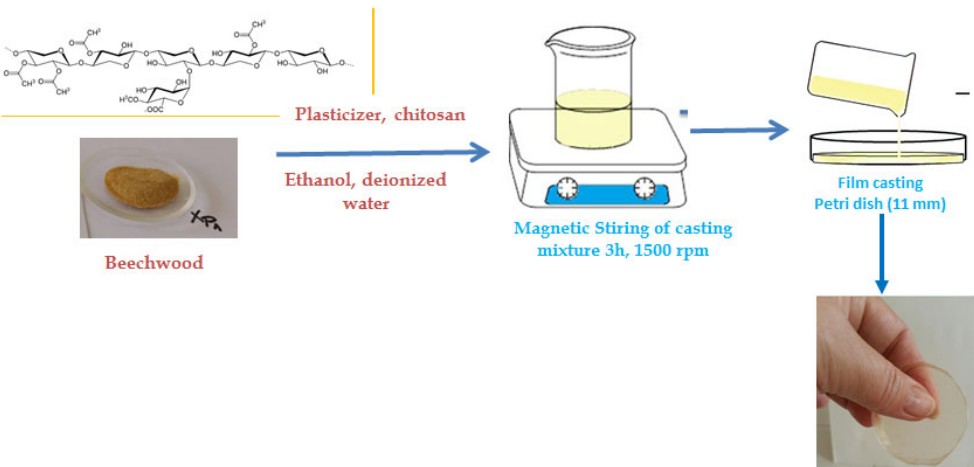

**Figure 2.** Preparation of xylan hemicellulose films.

- Surface Treatments (Coating) of Paper

The coatings were applied as thin layers of 0.75–1 g/m$^2$ on each side of the paper using the Mayer rod laboratory coating system. In this system, the dispersion of xylan was applied in front of the rod, and by manual rotation of the rod over the paper substrate, a well-defined amount of coating dispersion was applied. The thickness of the coating layer was controlled by the diameter of the wire. The obtained samples were dried for 10 min at room temperature and 10 min in an oven at 60 °C.

2.2.1. Testing Methods for Coated Paper Samples

- Air Permeability

This was evaluated as a measure of time (s) for the passing of air volume through paper samples, with settled area according to ISO 5636-5:2013 by the Gurley method.

- Water Absorption Capacity

This was determined as described in the standard method SR ISO 535:2014, where a given amount of water was in contact with the paper for 60 s and weight differences were compared (Cobb60 index).

- Oil Absorption Capacity

This was measured according to the T-441 om-98 standard, where a given amount of rapeseed oil was in contact with the paper for 600 s, and weight differences were compared (Unger-Cobb600 index).

- Water Vapour Permeability (WVTR), g/m2.day

The measure of water vapour transmission rate was evaluated according to SR EN ISO 15106-1:2005 Part 1: Method with humidity detector; it was determined by measuring the time necessary for the increase in humidity in the top chamber, from a predefined minimum value to a predefined maximum value. The measured time was compared with the time registered in the calibration process of the standard film with known permeability, and the result was expressed as water vapour transmission rate in g/m$^2$/24 h.

- Static Contact Angle (CA)

CA tests were performed by the sessile drop method, according to the TAPPI T 458 cm-04 standard (2004), using a Kyowa goniometer, Model DM-CE1, equipped with a digital camera and software for recording and processing results. Paper samples were fixed with clamps on the goniometer test table, and then water drops were deposited on its surface with a microsyringe. The value of the contact angle was recorded after a water–substrate contact time of 5 s on samples. A total of 10 measurements were taken for each sample (Figure 3).

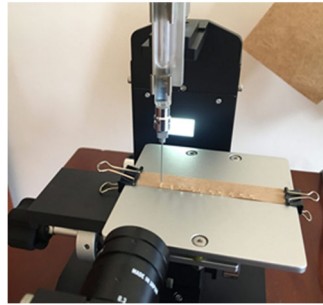 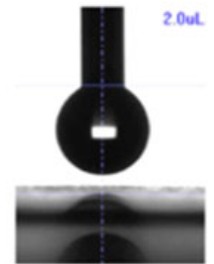 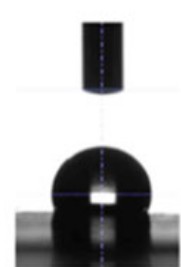

**Figure 3.** Kyowa goniometer for contact angle measurements.

- Dry and Wet Tensile Strength

This was measured using Instron extensometer, as maximum tensile strength on width length, which was supported by paper sample until breaking point, according to SR EN ISO 1924-2:2009 and ISO 3781:201.

- Tearing Strength

This is the tearing force that continues an initial cutting within one paper sample, according to SR EN ISO 1974:2012; the Elmendorf tearing strength tester was used for this determination.

- Bursting Strength

This is an important property for packaging papers and is determined by the maximum hydrostatic pressure required to rupture the sample of paper. It was measured using Lorentzen and Wetre bursting strength tester, according to SR EN ISO 2758:2015.

- Antimicrobial Activity

A qualitative screening was performed using an adapted agar diffusion technique: Petri dishes containing Muller–Hinton solid medium were inoculated with a 1 mL bacterial suspension, obtained from pathogenic cultures of 24 h on Muller–Hinton broth medium; then, pieces about 1 cm$^2$ area from samples of coated and reference paper were placed on solidified media inoculated with *Escherichia coli* and *Staphylococcus aureus* bacteria. The resulting plates were incubated for 24 h at 37 °C. The appearance of a growth inhibitory zone was interpreted as an antibacterial effect. The inhibition zones were measured and expressed in mm.

- Antifungal Activity

The microbiological activity of xylan-coated papers was evaluated against standard fungi. The procedure consisted of the application of sterilised paper pieces (about 1 cm$^2$ area) on malt mould agar (MMA) culture medium placed in Petri dishes. The level of fungi growth was evaluated after 5–7 days of incubation at 20–25 °C. The fungi growth was identified by sample observation under a stereomicroscope. The captured images were analysed, and results were expressed as the percentage of paper surface covered by fungi.

2.2.2. Testing Methods for Xylan Film Samples

Water swelling capacity was determined after 1 h immersion of film sample in distilled water at room temperature. It is considered that this period of time is sufficient to reach the equilibrium state. After the time was reached, the excess of water was removed with filter paper. Before ($W_{dry\ initial}$) and after water swelling ($W_{wet\ final}$), the weight of film samples was measured. The initial dry weight of the samples ($W_{dry\ initial}$) was calculated considering its dryness after treatment in an air oven at 103 °C [34].

The swelling capacity of film samples was calculated by Equation (1).

$$\text{Swelling capacity, } \% = \frac{W_{wet\ final} - W_{dry\ initial}}{W_{dry\ initial}} \times 100 \tag{1}$$

Water solubility was determined as the percentage of dry matter of film which is solubilised after 1 h immersion in distilled water at 25 °C. The samples of films were dried in an oven for 2 h at 103 °C and weighted to obtain the initial dry weight ($W_{dry\ initial}$). Then, the samples were immersed in 50 mL of distilled water at room temperature and for 1 h. The remaining samples (waste) were filtered by filter paper and dried at 103 °C for 4 h and weighed to obtain the final weight ($W_{dry\ final}$). The solubility capacity of film samples was calculated by Equation (2) [35].

$$\text{Water solubility, } \% = \frac{W_{dry\ initial} - W_{dry\ final}}{W_{dry\ initial}} \times 100 \tag{2}$$

## 3. Results and Discussions

### 3.1. FT IR Analysis of Xylan Biopolymer

The analysis of xylan spectrum is shown in Figure 4, while the observed functional groups are presented in Table 2.

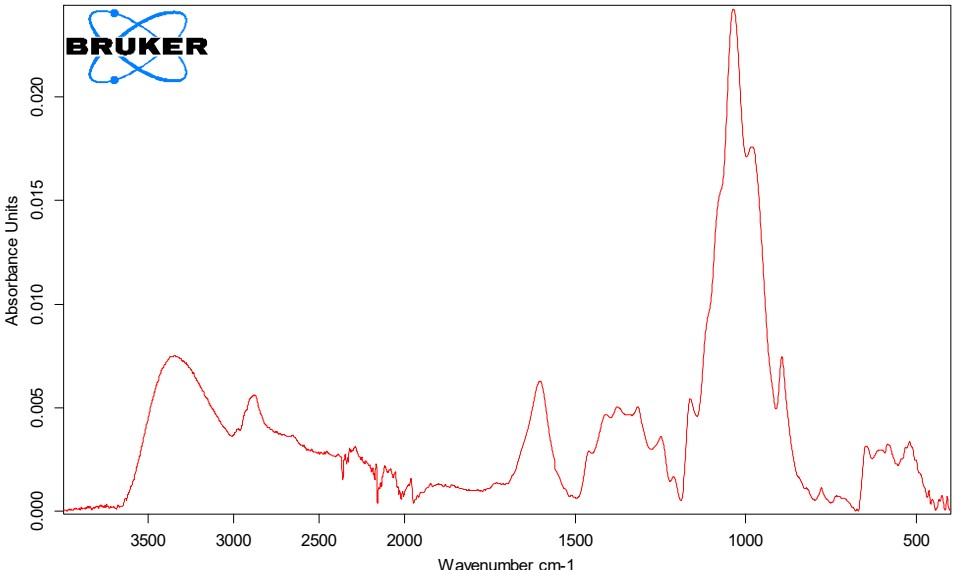

**Figure 4.** FTIR Spectrum of xylan hemicellulose.

**Table 2.** Structural evaluation of xylan hemicelluloses.

| Region/Area, cm$^{-1}$ | Stretching Vibration | Class of Potential Functional Groups |
|---|---|---|
| 1000–1200 | $\nu_{C\text{-}H\text{-}O}$/$\nu_{C\text{-}OH}$ | esters/ethers |
| 3000–3700 | | alcohols |
| ~3410 | $\nu_{CO\text{-}H}$ | aldehydes |
| 2800–3050 | $\nu_{C\text{-}H}$ | -CH$_3$, -CH$_2$, -CH |
| 1460 | $\nu_{\text{-}COCH3}$ | acetyl l |
| 2839 | $\nu_{\text{-}CH2}$ | metoxi |
| 2856 | $\nu_{\text{-}CH3}$ | C-CH$_3$ |
| 1725–1737 | $\nu_{c=o}$ | aldehyde/ketone groups |
| 1414 | $\nu_{coo^-}$ | carboxylic acids |
| 894 | 1–4 glycosidic bond between xylopyranose units | xylanic chains |

As it is observed in Table 2, a high number of hydroxyl groups in the structure of xylan hemicellulose are present, thus increasing its hydrophilic character. However, the high number of hydroxyl groups facilitates the chemical modification of xylan hemicelluloses when new functional groups can be introduced to improve its hydrophobicity.

### 3.2. Barrier Properties of Xylan-Coated Papers

Liquid absorption and gas/vapour permeability are the most important properties required for packaging papers that come into contact, temporarily or permanently, with aqueous liquids and wet foods. Current solutions to obtain adequate barriers for food packaging paper are coatings with synthetic polymers/waxes and lamination with plastic or aluminium foils [36]. Literature reviews show the main applications of hemicelluloses as a barrier material for stand-alone packaging films, and only a few of them are related to packaging paper coatings [37].

As it can be observed from the results presented in Table 3, in comparison with the base paper, xylan-coated samples exhibit a reduction in air volume and water vapours that pass through the paper structure.

**Table 3.** The water and air barrier properties of xylan-coated papers.

| Property | Base Paper | Xylan Coated Paper |
|---|---|---|
| Air permeability, Gurley, s | 89.95 | 167.1 |
| Water vapours transmission rate, g/m$^2$/day | 233.3 | 211.63 |

This can be explained as follows: due to their hydrophilic properties, xylan hemicellulose blocks the air through the cellulose fibre network very effectively; in addition, based on its ability to form uniform films, xylan is able to fill and close the pores on the paper surface [38,39]. This is confirmed by the improvement in water and oil absorption of coated paper samples (Figure 5).

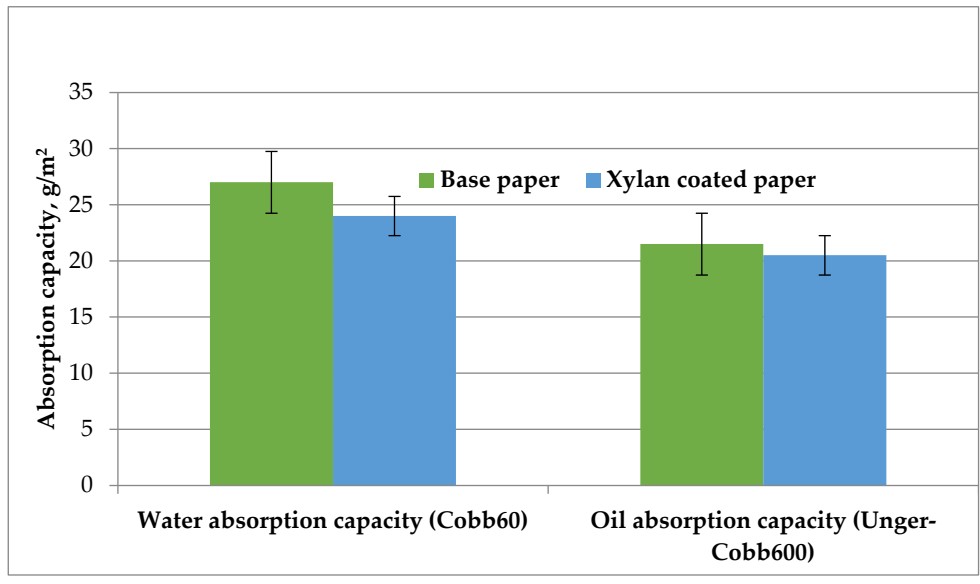

**Figure 5.** The liquids absorption capacity of xylan-coated paper samples.

The obtained results for air permeability, water absorption, and WVTR of xylan-coated paper samples can be compared with those obtained by Manea et al. (2011) for food packaging paper treated with fluorochemicals (Lodine 2000). Under similar conditions, they obtained greaseproof paper (55 g/m$^2$) with an air permeability of 158 s, water absorption capacity of 22 g/m$^2$, and water vapour transmission rate (WVTR) of 198.11 g/m$^2$/day, respectively [40]. In addition, in their research, Anthony et al. (2015) have tested hemicelluloses as binders in paper coatings in comparison with polyvinyl alcohol (PVA) coatings. The results indicated that water absorption and water vapour transmission rate (WVTR) for both PVA- and hemicelluloses-based coatings were similar [41]. In other studies [42], the xylan hemicellulose was used as a binder in pigmentary coatings based on kaolin and calcium carbonate. The coated papers were compared with those coated with acrylic latex (Styronal 302) as the binder. The air permeability of coated paper with xylan binder coatings was slightly higher than values obtained for coatings with acrylic latex, at 10 pph. By using corn cob xylan for surface treatment of linerboard, Witherspoon (2018) obtained an improvement in gas permeability and a slight decrease in mechanical strength of coated papers, at 15 g/m$^2$ weight of coating layer [43].

Contact angle (CA) is a measure of the capacity of fluids to adhere and wet the surface of different substrates. It is a widely used technique for studying the loss and recovery of hydrophobicity of polymer films. For coated papers, wetting is a complex procedure involving the spread and absorption of water into the coating layer structure. Absorption starts after the drops have wetted the surface to a certain extent.

Analysing the results presented in Figures 6–8, it can be seen that the contact angle values for xylan-coated paper samples do not vary significantly, compared with the paper

substrate. The value of water absorption for the base paper indicates that this substrate is medium sized. This limits the penetration of xylan dispersion within a fibrous network. As a result, a higher amount of xylan polymer remains at the paper surface which is hydrophilic and hydrosoluble [39]. In this case, the contact angle of the coated samples is influenced by the nature of the coating layer only.

Generally, hemicellulose-based coatings are hygroscopic and absorb moisture. This is because the hemicelluloses have abundant free hydroxyl groups with a high affinity to water distributed along the main and side chains [44]. Furthermore, the formation of the composite film reduced the direct contact area between the paper fibres and water and weakened the combined effects. This leads to an increase in the fibre's attraction to water [45].

The obtained CA values are similar to those reported by Anthony and collab (2015), who obtained coated papers with CA of about 112° using native hemicelluloses with 26% content of xylan (about $4 \, g/m^2$), extracted by distiller's grains. These values were higher than CA of coated papers with polyvinyl alcohol dispersions at the same coating weight. Moreover, the comparative values of CA have been reported by Laine et al. (2013) by the replacement of acrylic latex with modified xylan in coatings for paper packaging [41,42].

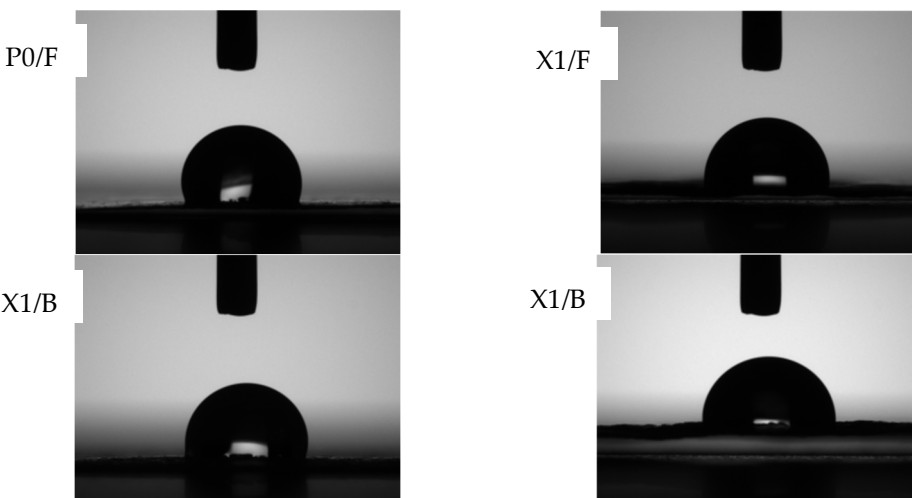

**Figure 6.** Images of water drop on the coated paper samples surface. P0, base paper; X1, xylan-coated paper sample; F/B, front side/backside of sample.

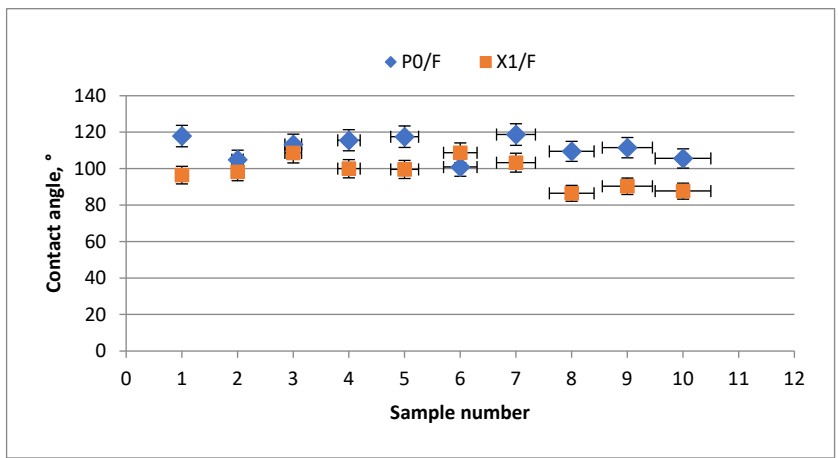

**Figure 7.** The values of contact angle for base paper and xylan-coated paper samples (F, front size).

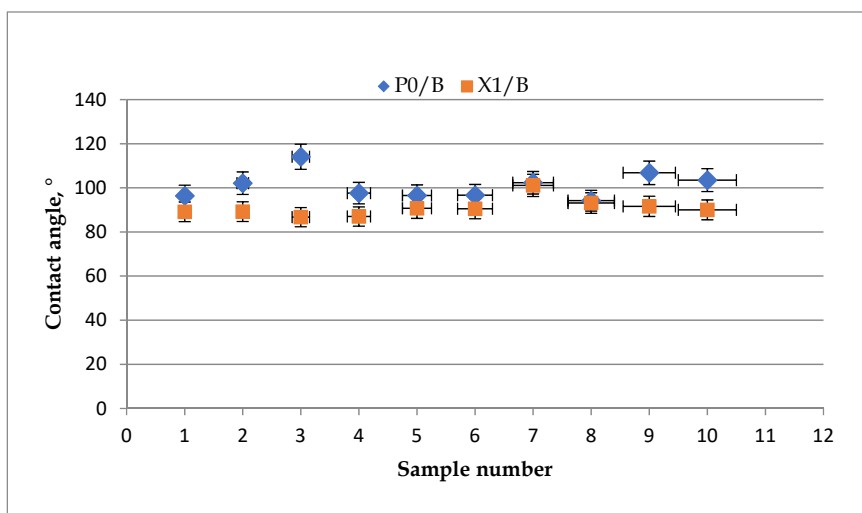

**Figure 8.** The values of contact angle for base paper and xylan-coated paper samples (B, back size).

### 3.3. Mechanical Strength Properties of Coated Paper Samples

In addition to functional properties, mechanical strength is very important for paper packaging to assure the appropriate protection of the packaged material during transport and handling. Moreover, for papers that will be used under wet conditions, high wet strength is preferred. The wet strength of the samples is expressed in terms of the wet/dry strength ratio, which is the wet tensile strength expressed as a percentage of dry tensile strength. Analysing the obtained results (Figure 9), it is observed an insignificant increase in the wet tensile strength of xylan-coated papers, in comparison with the base paper. This suggests that hydrogen bondings are still the predominant basis of the structure [38]. The decrease in dry tensile strength of xylan-coated papers may be suspected by the fact that xylan has lower mechanical strength than the paper substrate, which would lead to a decrease in tensile strength in the combined material. It is also possible that the xylan coating interferes with the hydrogen bond network, which reduces the tensile strength of the base paper as well.

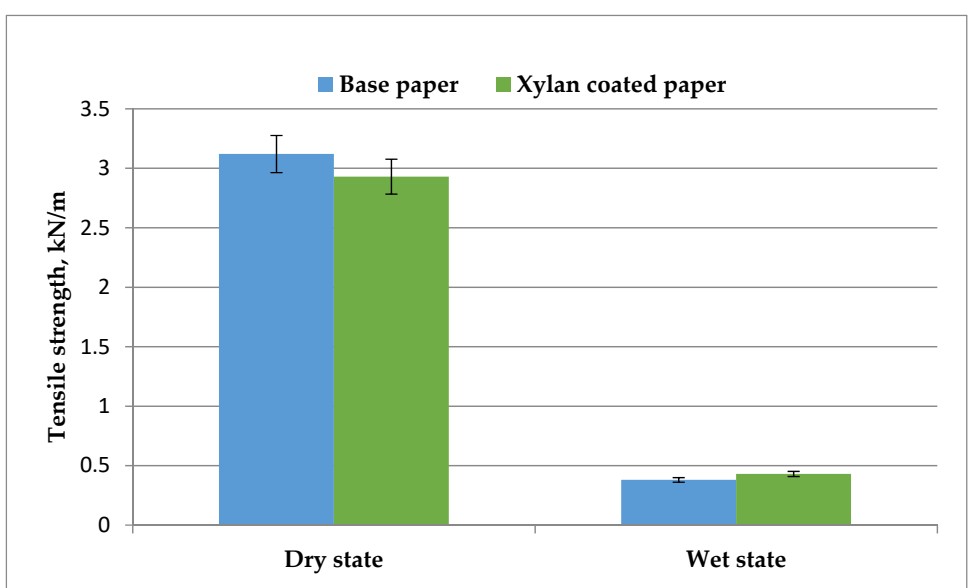

**Figure 9.** Tensile strength of xylan-coated paper samples.

The decrease in tearing strength (Table 4) is due to the fact that, as a result of coating treatments, the paper structure is more compacted (confirmed by a decrease in porosity), and the tearing force acts on a lower surface.

**Table 4.** The mechanical strength of xylan-coated paper samples.

| Property | Base Paper | Xylan Coated Paper |
| --- | --- | --- |
| Tearing strength, mN | 480 | 440 |
| Bursting strength, kPa | 200 | 279 |

As it is observed in Table 4, xylan treatment leads to the improvement in the bursting strength of coated paper. This can be explained by the fact that at low coating weights, the additional moisture in the paper substrate as an effect of xylan hygroscopicity is less and have no negative influence on the bursting strength of coated paper. Similar results were reported by Witherspoon (2018), who obtained an improvement in the bursting strength of filter paper coated with corn cob xylan at low coating weights [43].

*3.4. Antimicrobial and Antifungal Activity of Coated Paper Samples*

The antimicrobial activity of food packaging paper is of great importance in order to preserve food quality and extend shelf life; in recent years, this topic was intensively studied for lignocellulosic materials.

Inhibition zone is the most-used method for antimicrobial activity tests of materials. By using this method, bacterial growth is inhibited by the formation of a transparent zone resulting from the diffusion of antimicrobial agents in agar plates. As can be observed in Figure 10, there is no inhibition zone around the paper piece, for both base paper and coated samples with xylan hemicellulose.

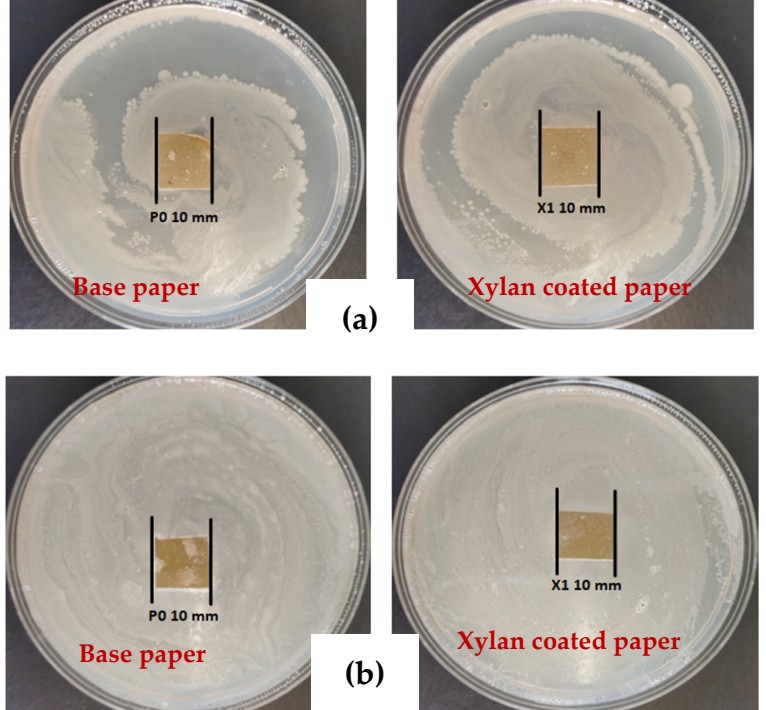

**Figure 10.** The antimicrobial activity of coated paper samples: (**a**) *E. coli*; (**b**) *S. aureus*.

However, a slight antibacterial effect of coated paper samples can be observed on its surface, where the number of bacterial colonies is lower than the reference sample, especially for *S. aureus* pathogenic microorganisms (Figure 10b).

Concerning the antifungal activity of xylan-coated papers, the inhibition effect of fungal growth was expressed as a percentage of paper surface covered with fungi. Therefore, while the surface of the uncoated paper was covered with fungi more than 35%, on the surface of xylan-coated papers, a very low quantity of spores, about 1.5%, was developed (Figure 11). In this case, the film-forming ability of xylan hemicelluloses contributes to the development of some mechanisms which can prevent the growth of spores on the paper surface [46,47].

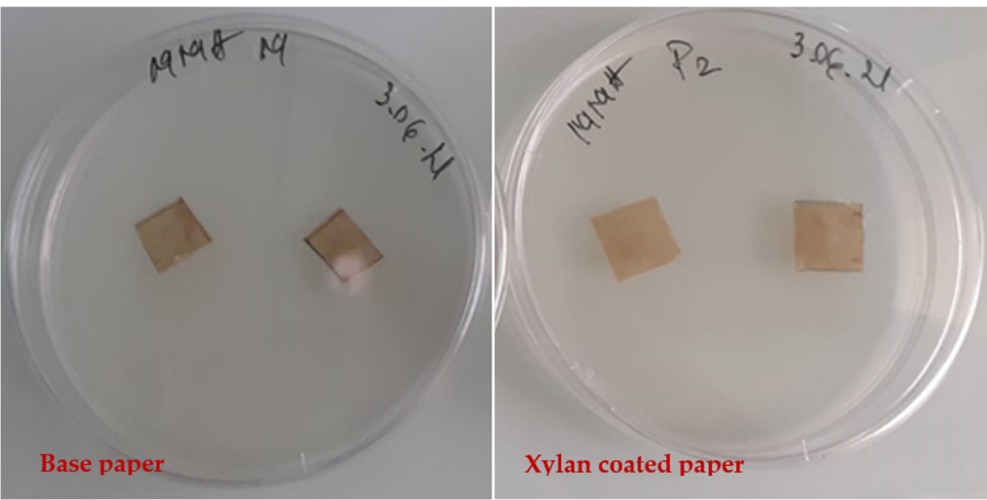

**Figure 11.** The antifungal activity of xylan-coated papers.

The obtained results are in accordance with the literature data regarding the antimicrobial activity of native hemicelluloses. Thus, in recent studies [34,48,49], the antimicrobial properties of paper coated with xylan hemicelluloses and films were reported. It was concluded that, in its native form, xylan shows a slight antimicrobial activity against pathogen bacteria (*E. coli*, *S. aureus*). In these studies, a substantial improvement in the antimicrobial activity of xylan hemicelluloses is obtained by chemical modification using esterification or crosslinking methods. In other research [50,51], it is reported that gels using xylan, gelatin glycerol, and nicotinamide have good microbiological activity against yeasts and fungi.

### 3.5. Film-Forming Ability of Xylan Hemicellulose and Water Barrier Properties of Films

Generally, the native xylan exhibits poor film-forming ability and forms high brittle films with very low mechanical stability. This is a direct consequence of the insufficient chain length of the polymer and poor solubility [12]. Commonly, plasticisers (sorbitol, xylitol, glycerol and propylene glycol, etc.) are added, or other biopolymers (wheat gluten, carboxyl methylcellulose, nanofibrillated cellulose, chitosan, etc.) are compounded along with xylan hemicellulose to enhance film forming and other properties of these films.

The capacity of water resistance is the most important property for food packaging materials. The swelling capacity showed by all films is due to the high hydrophilicity of xylan and chitosan polymers [34]. In this study, the chitosan biopolymer and glycerol plasticiser were used to improve the properties of xylan films (50 XCh, 70XCh, and 80XCh samples; Figure 12). In Table 5, t is shown that the mean value of swelling capacity of films with major xylan content is higher, probably due to the high hydrophilicity of hemicelluloses.

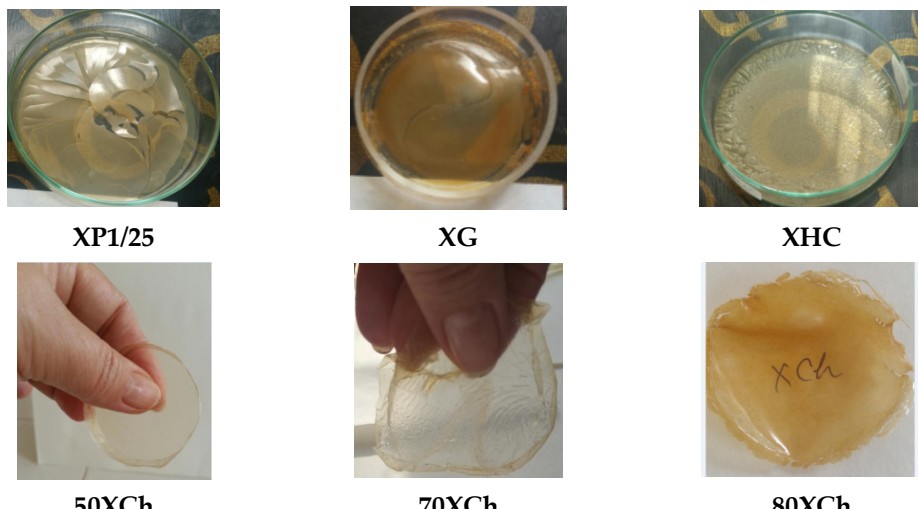

| XP1/25 | XG | XHC |
|--------|----|----|
| 50XCh | 70XCh | 80XCh |

**Figure 12.** The images of xylan hemicellulose films.

**Table 5.** The water barrier properties of xylan hemicellulose films.

| Sample | Swelling Capacity, % | Solubility, % |
|--------|---------------------|---------------|
| 50XCh | 73.2 | 78.3 |
| 70XCh | 85.4 | 82.4 |
| 80XCh | 90.0 | 95.5 |
| XHc | 94.2 | 99.23 |
| XG | 93.2 | 98.9 |
| XP1/25 (alcohol) | 93.5 | 99.25 |

It is important to mention that, in the swollen state, films with xylan and chitosan preserved their integrity, i.e., they were easily handled (Figure 13). This is beneficial for predicting their potential application for hydrogels and packaging purposes.

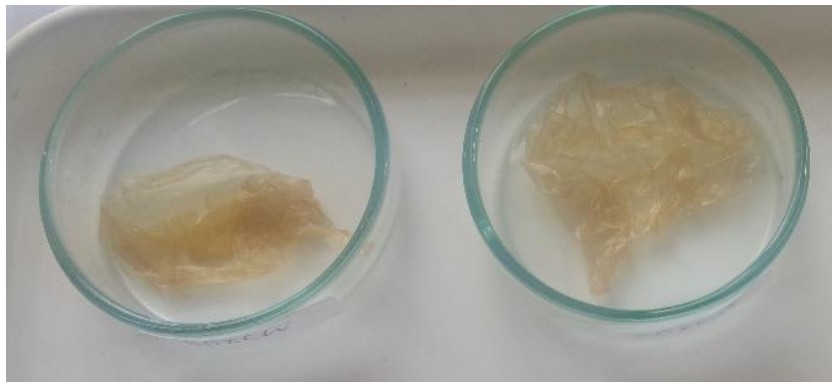

**Figure 13.** Xylan hemicellulose films after 1 h immersion in water.

The films' solubility is a relevant characteristic of packaging to improve and protect the packaged product integrity. It can be observed that there is an increase in solubility for films with high content of xylan. The presence of chitosan may reduce water absorption due to its low solubility in water. Due to the high hydrophilic character of xylan hemicellulose, the solubility of xylan films is higher than other polymeric materials used in packaging applications [35]. This can be decreased by the chemical modification of xylan biopolymers or by the addition of other compounds that are more hydrophobic [12,35,36].

## 4. Conclusions

This research study aimed to evaluate the performances of wood xylan polysaccharides in coating applications for paper food packaging and polyelectrolyte complexes formula for uniform films formation.

According to the obtained results, it can be concluded that by coating the paper with xylan hemicellulose, an improvement of water/oil and gas barrier properties was obtained. The xylan treatment had a positive effect on the bursting strength of the coated papers, which is one of the most important strength properties of packaging paper.

Measurements of microbiological activity revealed that the xylan-coated papers exhibited moderate antifungal activity against standard fungi and only a slight antibacterial effect against *S. aureus* pathogenic bacteria.

The film-forming ability of xylan hemicelluloses, as well as the swelling and solubility of films, can be improved by adding chitosan biopolymer. In swollen state, the polyelectrolyte complexes in xylan–chitosan films kept their integrity. This can predict their potential utilisation in packaging applications.

Although there is a limited number of research studies concerning the application of hemicelluloses in paper coatings, the obtained results reveal a promising starting point in anticipating the use of these biopolymers in food packaging applications, as competitive and sustainable alternatives to petroleum-based materials (i.e., fluorochemicals additives), to obtain environmentally friendly products according to a circular economy.

These preliminary results will help to design the future research directions, which will be focused on finding the appropriate pathways for chemical modification of xylan hemicelluloses by esterification or crosslinking with other compounds to reduce their hydrophilic character and to be more appropriate for paper coatings.

**Author Contributions:** Conceptualisation, P.N. and R.M.; methodology, P.N. and R.M.; validation P.N., R.M. and F.C.; investigations, P.N., R.M. and F.C.; writing—original draft preparation, R.M.; writing—review and editing, P.N.; visualisation, P.N.; supervision, P.N. All authors have read and agreed to the published version of the manuscript.

**Funding:** This research was funded by DUNAREA DE JOS UNIVERSITY OF GALATI, ROMANIA, grant number RF 3646/2021.

**Institutional Review Board Statement:** Not applicable.

**Informed Consent Statement:** Not applicable.

**Data Availability Statement:** Data are provided in the paper.

**Acknowledgments:** The authors are thankful for the support of the Research Centre for Environmental and Agriculture "*Lunca*", within Engineering and Agronomy Faculty in Brăila, "Dunărea de Jos" University of Galați, Romania.

**Conflicts of Interest:** The authors declare no conflict of interest.

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
