# Peer review of "Xylan Hemicellulose: A Renewable Material with Potential Properties for Food Packaging Applications"

_sustainability, doi:10.3390/su132413504_

Round 1

Reviewer 1 Report

The article "Xylan hemicellulose a renewable material with potential properties for food packaging applications" will be of great interest to potential readers. However, the authors will need to capture the essence of this application to the overall concept of a circular economy especially in the introductory comments (L34-37).

In order to improve the readability of the article, there is a need to revise the English grammar professionally. I have attempted to make some improvement to the language as below.

L41: Hemicelluloses (HCs) as t introduced for the first time here.

L42: ...and they represent an..

L45: pulp & paper; L46:..most of the hemicellulose remain in the fibers,...

L53: ......, the amounts vary as a function. of biomass sources, e.g. as..

L56: When compared to cellulose which occurs in the cell walls..

L70: and hexoses (C6H10O5) respectively.

L71-72: Besides these regular sugars, also exists their acidified forms in hemicelluloses, for instance ......

L78:...used to extract HCs are alkaline, hot water extractions, steam extraction..

Before 3.1. Xylan - the major types of hemicelluloses

It might be a good idea to introduce the readers to a short sentence on glucomannan, and mention that the main focus will be on xylan.

L90-93: Please revise the sentence for clarity

L99-100: ...and biofuels obtained from the biological conversion of sugar,..... Xylitol is the most widely produced from xylan, it is obtained by chemical...

L105-110: Please revise the sentence for clarity

L116: ....for film forming can be obtained.

L118: This will broaden the application areas of hemicelluloses

L123-126: ...therefore have the potential for application on food paper packaging especially in (oxygen-sensitive dairy products......). Furthermore,....., it can prevent the migration of mineral oils...

L171: This was evaluated.... according to ISO...by Gurley method.

L174: This was determined...

L178: This was measured by..

L182: The measure of water vapours...

L197: This was measured..according to SR EN ISO...

L199-200: This is the tearing force..  according to SR EN ISO...

L203-204: This is an important....according to SR EN ISO...

L240-243: Please revise the sentence for clarity

L246: ....that come into contact...

L247-249: ...in table 6, the xylan coated......air volume....paper structure in comparison to the base paper. This can be explained as follows:

L261: ...indicator in understanding...

L262: Due to the hydrophobic nature of hemicellulose.

L266: Do you mean: ...and its affinity to water?

L277: It is also observed...

L281-283: ....that as a result of........(confirmed by a decrease in porosity...)..The increase in bursting...

L319: ..to keep their integrity...This can predict their potential application for...

L324: It can be observed that there is an increase in solubility for...

L326: Due to the high hydrophylic character of..

L329-330: or by the addition of other compounds that are more hydrophobic.

L346: ...by esterification...

Consider adding any limitation of this study.

Author Response

Thank you very much for your useful observations, comments and recommendations. We deeply appreciate your evaluation.Your fruitful comments helped us to improve the overall quality of the paper.

Please, find attached, point by point, the response at your comments.

Reviewer 2 Report

The submitted article presents a very interesting idea. Nevertheless, the manuscript seems more like a work in progress and is not ready for publication yet. There is an extended use of tables in the materials and methods section that could be well substituted by brief description and supply references. In addition, the authors must decide on the abbreviations description for XyHCs or HCs for the xylans references within the text.

A key idea is “…that these materials can be used in food packaging applications as competitive and sustainable alternative to the petroleum-based biopolymers”. Unfortunately, no petroleum-based biopolymers were tested. As a result, the article lacks balance and seems asymmetrical, if I may express it that way. A series of tests were the xylan coating proposed is compared to actual additives and/or preservatives used commercially under the same conditions would be of great service to their case. On the other side a clear statement of the properties of packaging materials targeted is recommended.  Most figures shown do not clearly demonstrate a difference in properties from the Paper base to the coated version, and the evidence for antimicrobial activity is not conclusive. The scientific language in English could benefit from a third-party revision, especially from a native speaker.

The idea is of great impact and interest not only to the scientific community, but to the industrial and economy professionals as well. The authors should resubmit a thoroughly revised version of their interesting work.

Author Response

Thank you very much for your useful observations, comments and recommendations. We deeply appreciate your evaluation.Your fruitful comments helped us to improve the overall quality of the paper.

Please, find bellow, point by point, the response at your comments:

Reviewer 3 Report

Dear Authors,

My comments bellow:

Lines 188-196: Could you explain please how did you measure/ figured out the contact angle values?
Lines 262-264: Only afer a SEM observations we are able to say the hydrophilic and hydrophobic structures are not linked together - are to separate states. Without these observation you may just thinking the structures are not mixed together. So please rewrite this sentence. 
Why do you mention the energies if values are not presented?
Lines 312-313: I would rather say one of the most important, because some gases also play sgnificant role.
Overal: The explanation of obtained results should be deeper, lack of comparisons to the other authors, methods are only generaly described, the conclusions are mostly not about presented work. Please focus on presentation of your results deeply in the last section.

Author Response

Dear reviewer,

Thank you very much for your useful observations, comments and recommendations. We deeply appreciate your evaluation.Your fruitful comments helped us to improve the overall quality of the paper.

Please, find attached, point by point, the response at your comments.

Round 2

Reviewer 1 Report

The authors have improved the content of the manuscript and addressed most of my comments. However, it will be important to revise the English language of the article. This is entirely up to the journal if they handle the revision after the paper is finally accepted. 

L427-428: Do you mean - "modification of xylan hemicelluloses by esterification or crosslinking with other..."?

Reviewer 2 Report

An extended editing has been carried out. Some typing errors persist; nevertheless, this manuscript is an improved version almost ready to be published. I suggest strengthening of the antifungal argument. For instance, the evaluation could be accompanied with an area (percentage) of material free or invaded by the fungus and compared to paper control to support objectively and quantitatively the argument. The latter is new information and requires major revision. Please have a final proofreading by a person whose native to English.

Reviewer 3 Report

-

Author Response

Thank you for your reviewing.